# Can holographic optical storage displace Hard Disk Drives?
Jiaqi Chu ⊠, Nathanaël Cheriere, Grace Brennan , Mengyang Yang, Greg O'Shea, Jannes Gladrow, Douglas J. Kelly, Giorgio Maltese, Alan Sanders, Dushyanth Narayanan , Benn Thomsen & Antony Rowstron

Cloud data workloads require both high capacity at low cost and high access rates. Hard Disk Drives are the dominant media in this application as they are low cost, however, Hard Disk Drive technology is seeing declining access rates and a slowdown in capacity scaling. Holographic data storage could disrupt Hard Disk Drives in the cloud since it may offer both high capacity and access rates. However, a challenge with rewritable holographic media is the data durability due to erasure. Here we present a media and workload aware energy optimization framework and show that erasure can be managed. We investigated the optimal Fe concentrations in iron-doped lithium niobate with experimental results supporting a stretched-exponential erasure model. We achieved a record number of reads, and surpassed the previous record for density. Our approach provides an objective assessment the feasibility of such storage technology given component parameters and material properties.

The dramatic growth in cloud computing and the applications that this infrastructure supports are challenging the existing underlying data storage technologies. The applications that a particular storage technology can address in the cloud depends on two key metrics: the storage capacity per $ expressed as GB/$ and the access rate performance measured in read and write operations (IO) per second per unit of storage – IOPS/TB. In Fig. 1a we show where the existing cloud data storage technologies: Solid-State Drives (SSDs), Hard Disk Drives (HDDs), fit on this landscape. Broadly speaking, SSDs are used to support applications that require high access rates (known as hotter workloads), the more cost-effective HDDs are used for applications, such as email, that require large storage capacities and modest access rates (warm workloads). Magnetic tape is used to store archival or backup data that is seldom accessed (cold workloads).

In this work, we are focusing on the warm segment of this landscape where HDDs, which have been around for over 40 years, still support the bulk of the storage in the cloud today as they provide the most cost-effective storage medium with acceptable access rates. However, HDD technology is now approaching its physical storage density limits, with single drive capacity likely to top out at around 120 TB[1,2]. In addition, HDD access rates measured in IOPS, are fundamentally limited by the mechanics of spinning disk media (as shown by Fig. 1b) and have not increased for many years[3]. The consequence of capacity scaling via increasing the areal density and constant IOPS is that IOPS/TB has been declining. In the Cloud this decline is a challenge because HDDs are no longer able to provide the required access rates for the random read operations that dominate in these applications. The current solution to this problem is to offload the hotter parts of the storage workload to SSDs which offer much greater IOPS/TB but cost approximately 6–9× more per TB of capacity. The demand for an alternative technology that can fit in the warm data space with density and capacity that is competitive with HDD, but better access rates at low cost has become urgent.

Holographic data storage (Fig. 1c) may be suitable because it leverages the volume of the media when writing and reading pages of bits, instead of just accessing the surface of the storage media. The theoretical density limit of $\lambda^{-3}$ is promising, and the page-wise storage mechanism seems destined for optimized throughput. The interest in holographic data storage drove intensive research in the 1990s through to the early 2000s. The photorefractive effect and the physical mechanisms are now well understood[4,5]. In demonstrations of volatile storage systems, a density up to 1.08% of the theoretical limit[6] was reported. However, most of the research efforts were ceased twenty years ago, partly because the experimentally achieved capacity was below the expectations set by HDD[7] while HDD capacity was still growing exponentially at that time. Another reason was seemingly the lack of ideal holographic storage media that is both durable and efficient[8]. Given the saturation in the areal density scaling for HDDs and the associated decline in IOPS TB$^{-1}$ causing challenges for servicing the access rates of workloads in the cloud era, it is timely to look again at holographic storage media to see if it could provide cost-effective density whilst increasing the IOPS TB$^{-1}$.

In this work we focus on holographic storage in rewritable photorefractive Fe:LiNbO$_3$ crystal that is suitable for warm data storage, in contrast to write once holographic media such as photopolymers[9] that are only suitable for archival storage. We have developed an end-to-end framework (Fig. 2a) considering material properties, storage system workloads, and

Microsoft, 198 Science Park, Milton Road, Cambridge CB4 0AB, UK. ⊠e-mail: jiaqchu@microsoft.com

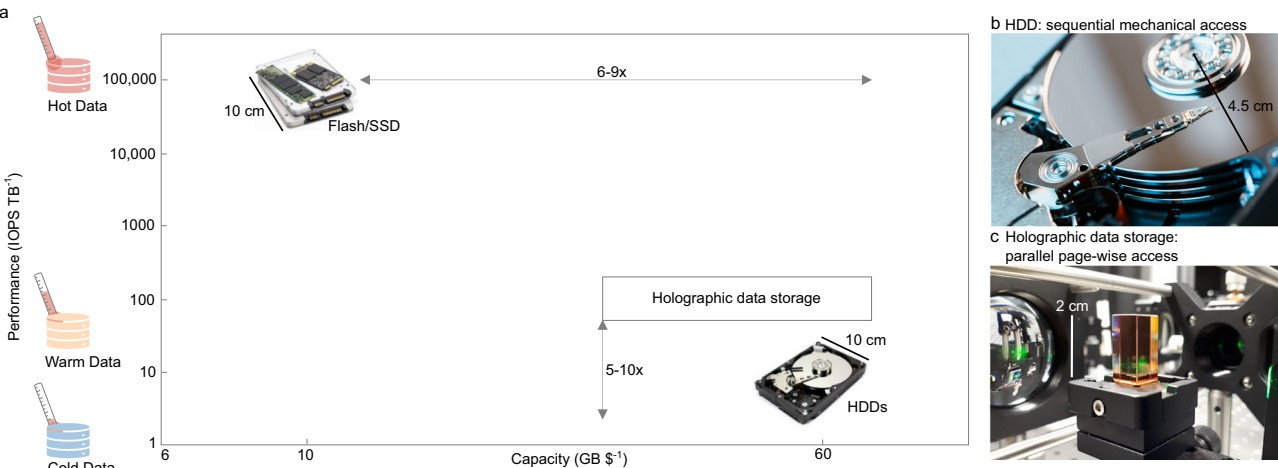

**Fig. 1 | Quantifying the cloud storage landscape in terms of the cost (GB \$$^{-1}$) and the access rate performance (IOPS TB$^{-1}$).** IOPS stands for input/output per second. SSD stands for Solid-State Drive. HDD stands for Hard Disk Drive. **a** Performance of existing technologies highlighting the warm data segment between 10 and 100 IOPS TB$^{-1}$ that Holographic storage technologies are aiming to target. **b** Schematic of spinning mechanics in HDDs. **c** A LiNbO$_3$:Fe crystal that leverages 3D data storage. Image components of SDD and HDD courtesy of Shutterstock, used under license.

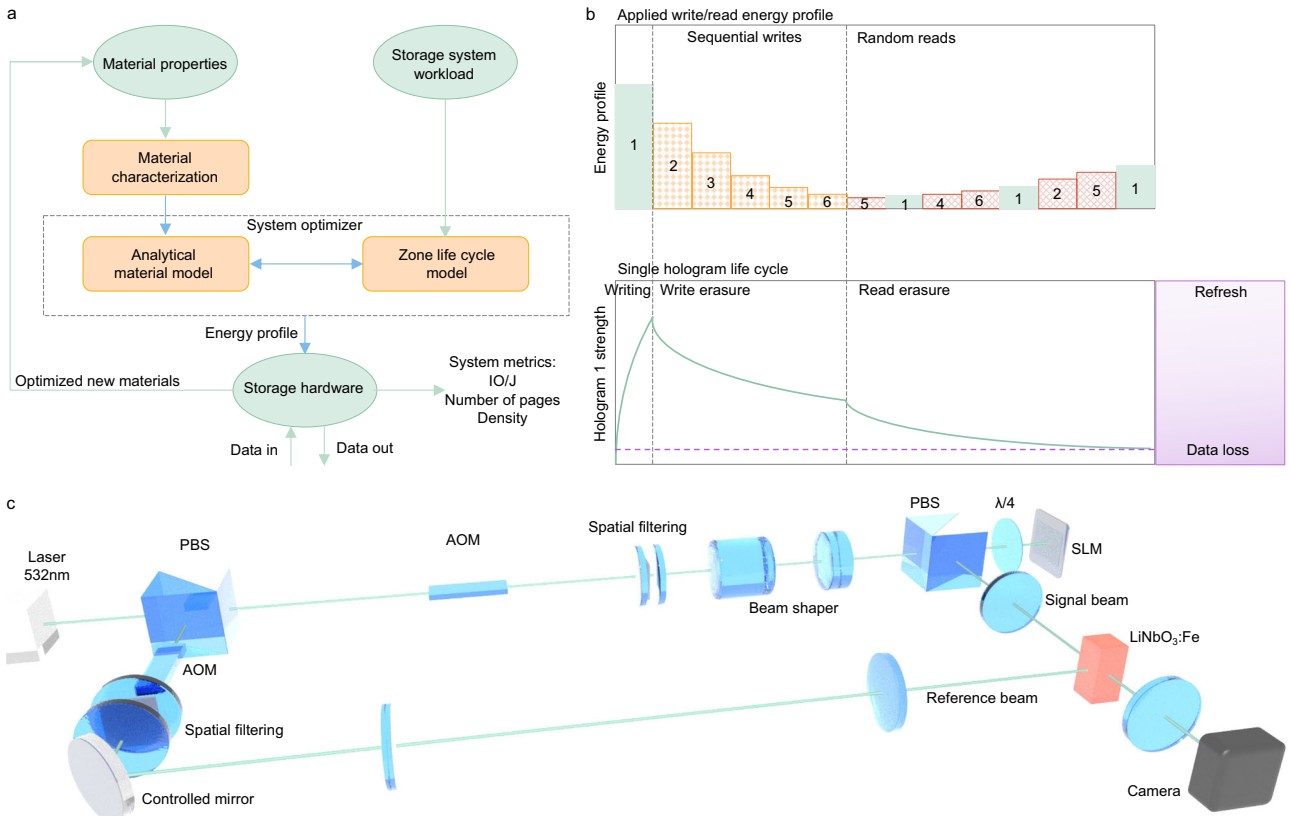

**Fig. 2 | An end-to-end framework that quantifies and optimizes energy efficiency and density. a** Schematic of the end-to-end zone lifecycle optimizer. **b** Illustration of zone lifecycle. The top graph provides a schematic of write and read energies, which are used to do sequential hologram writing and random hologram reading. The lower graph provides a schematic of how the strength of the first hologram changes when sequential writes and random read are occurring in the same zone. Holograms degrade every time there is a write or read, so there is a point (e.g. marked by the horizontal dashed line) after which the data written in a zone becomes unreadable. **c** Schematic of the storage hardware rig used to validate end-to-end performance. PBS polarized beam splitter, AOM acousto-optic modulator, SLM spatial light modulator. λ/4 quarter-wave plate.

storage hardware implementation to assess the entire storage system. Experimental material characterization determines the write and erasure properties of the media as the function of the Fe doping and annealing. These properties are then combined with workload-aware system model to optimize the energy profile to a specific workload. We then validate these profiles in a storage hardware rig that stores and reads out data from the media. Based on the findings, we can select optimized materials for cloud data storage. This allows us to quantify the trade-off between key system metrics, energy efficiency and the density. We use these two physical metrics as they are a good proxy for the key storage system metrics \$ GB$^{-1}$ and IOPS TB$^{-1}$. The cost of a

holographic storage device is proportional to the energy per operation / storage density whilst the IOPS TB$^{-1}$ are proportional to the energy per operation. Thus, increasing both the energy efficiency per operation and density is essential to achieving the required IOPS TB$^{-1}$ whilst minimising the cost per TB of the storage device. In contrast prior work focused solely on individual aspects such as dynamic range[10–12] or write efficiency[13–15].

A key challenge with holographic storage systems that employ rewritable photorefractive crystal media is read/write erasure, where subsequent write and read operations to the same zone degrade the fidelity of the stored pages in the zone as illustrated in Fig. 2b. This means that we need a mechanism to refresh the data before it becomes unreadable. Write and read erasure imposes challenges on achievable number of pages (thus, density) and energy efficiency. Previously, write erasure was mitigated by profiling the energy used to write each page[10]. We have refined the previous exponential erasure model with a more accurate stretched-exponential model, improving optimization of the write energy profile. Further we jointly optimize read/write energy profiles, tailoring them for workloads with varying ratio of read/write operations, as depicted by the system optimizer block in Fig. 2a. Following this protocol, we show that the data durability challenge can be tackled by implementing a refresh operation in tandem with garbage collection. We show a 1.9× improvement in best achievable energy efficiency at media, measured by the number of read/write operations (IO) per Joule of optical energy incident on the media (IOPS TB$^{-1}$), which is a good indicator of storage system access rate and cost. In simulation, this optimization also enables us to do 3.4× more reads at best energy efficiency before a refresh operation is required. In experiments with a setup shown by Fig. 2c, we demonstrated 392 net IO J$^{-1}$ at 400 pages for a workload with 100% reads (with a page size of 47KB and a raw density of 1.2 GB cm$^{-3}$ in a 0.015% Fe doped crystal with $1.2 \times 10^{17}$ cm$^{-3}$ Fe$^{2+}$). This equates to 9.3 K read operations before a refresh cycle is required. We demonstrated 185 net IO J$^{-1}$ at the same number of pages for a workload with 50% reads.

We have undertaken a demonstration of density where we established a benchmark for net density, achieving 9.6 GB cm$^{-3}$ with 705 multiplexed pages, by increasing the net page size to 73KB and improving data recovery using machine learning (note: the net density includes all the overheads e.g. error correction, the raw density here was 16.8 GB cm$^{-3}$). In this experiment, we achieved an energy efficiency of 108 IO J$^{-1}$ using a 0.03% Fe doped crystal with non-optimal Fe$^{2+}$ of $4.11 \times 10^{17}$ cm$^{-3}$.

Additionally, we have investigated the impact of the Fe doping and annealing conditions of Fe:LiNbO$_3$ media, as listed in Table 1, on the system performance. The crystals are labelled by their Fe doping percentage and the ratio of Fe$^{2+}$ to Fe$^{3+}$ concentration. We show that the optimum Fe doping and Fe$^{2+}$ concentration post annealing is as high as possible and $2.5 \times 10^{17}$ cm$^{-3}$, respectively. This is due to the dominant influence of Fe$^{2+}$ concentration on write efficiency and the ratio of Fe$^{2+}$ to Fe$^{3+}$ concentrations on erasure properties, which are both critical for energy profile generation. We show that the upper bound for Fe:LiNbO$_3$ with scalable number of pages is approximately 100 net IO J$^{-1}$. With this energy efficiency, our calculations indicate the possibility of achieving up to 2201 pages per zone for a workload with 100% reads, and 2910 pages for a workload with 50% reads. (This equates to raw densities of 6.6, 8.7 GB cm$^{-3}$, assuming a small page size of 47KB.) In experiments, we measured the write efficiency and erasure properties of a near optimal crystal with 0.02% Fe doping and $3.1 \times 10^{17}$ cm$^{-3}$ Fe$^{2+}$. The energy profiles generated from this analysis show an achievable count of 1482 pages per zone at 100 net IO J$^{-1}$ for a workload with 100% reads, and 1918 pages for a workload with 50% reads, corresponding to densities of 4.4, 5.7 GB cm$^{-3}$ assuming 47KB per page. This framework can be used to analyze performance of untested materials as well.

## Results
### Hologram dynamics: write and erasure
In the write phase, diffraction efficiency $\eta$ as a function of write fluence can be modelled as[5] (Fig. 3a)

$$\eta^{1/2} = A_s[1 - \exp(-F/\tau_r)] \tag{1}$$

**Table 1 | Fe:LiNbO$_3$ media sample properties**

| Label | Doping of Fe (mol%) | $c_{Fe}$ ($\times10^{18}$ cm$^{-3}$) | $c_{Fe2+}$ ($\times10^{18}$ cm$^{-3}$) | $c_{Fe3+}$ ($\times10^{18}$ cm$^{-3}$) | $c_{Fe2+}/c_{Fe}$ | $c_{Fe2+}/c_{Fe3+}$ |
|---|---|---|---|---|---|---|
| Fe0.001:r0.19 | 0.00105 | 0.198 | 0.032 | 0.166 | 0.161 | 0.192 |
| Fe0.001:r6.01 | 0.00105 | 0.198 | 0.170 | 0.028 | 0.857 | 6.008 |
| Fe0.002:r0.08 | 0.00214 | 0.404 | 0.030 | 0.375 | 0.073 | 0.079 |
| Fe0.002:r0.19 | 0.00214 | 0.404 | 0.064 | 0.341 | 0.157 | 0.186 |
| Fe0.002:r0.24 | 0.00214 | 0.404 | 0.079 | 0.326 | 0.195 | 0.242 |
| Fe0.002:r0.34 | 0.00214 | 0.404 | 0.103 | 0.302 | 0.254 | 0.340 |
| Fe0.002:r0.43 | 0.00214 | 0.404 | 0.122 | 0.282 | 0.301 | 0.433 |
| Fe0.002:r1.44 | 0.00214 | 0.404 | 0.239 | 0.166 | 0.590 | 1.438 |
| Fe0.005:r0.08 | 0.005 | 0.945 | 0.073 | 0.892 | 0.077 | 0.083 |
| Fe0.01:r0.04 | 0.01 | 1.89 | 0.065 | 1.825 | 0.035 | 0.036 |
| Fe0.01:r0.05 | 0.01 | 1.89 | 0.094 | 1.796 | 0.050 | 0.053 |
| Fe0.015:r0.04 | 0.015 | 2.835 | 0.117 | 2.718 | 0.041 | 0.043 |
| Fe0.02:r0.09 | 0.02 | 3.78 | 0.306 | 3.474 | 0.081 | 0.088 |
| Fe0.03:r0.08 | 0.03 | 5.67 | 0.411 | 5.259 | 0.073 | 0.078 |

Crystals are labels in the way Fe doping in %:ratio of Fe$^{2+}$ to Fe$^{3+}$ concentration.

where $A_s$ is the saturation diffraction efficiency, $F$ is the write fluence with a unit of cm$^2$ J$^{-1}$, $\tau_r$ is the write constant in unit of fluence. We have adopted this unit because the diffraction efficiency is contingent solely upon the light energy (in a defined area) for hologram write, rather than exposure duration, specifically in the low light intensity regime (Supplementary Note 1). We can also define the write efficiency $A_s/\tau_r$ that quantifies the rise in diffraction efficiency during the write phase because at $F << \tau_r$ for multiple page writing, $\sqrt{\eta} \propto A_s/\tau_r$.

As more holograms are written in the same area, the diffraction efficiency of previously written pages degrades with each subsequent write or read (Fig. 3b). In contrast to the widely adopted exponential decay[6], we find the decay is best described by a stretched-exponential model

$$\eta^{1/2} = \eta_0^{1/2} \exp[-(F/\tau_e)^\beta] \tag{2}$$

where $\eta_0$ is the initial diffraction efficiency, $F$ is the erasure fluence of subsequent holograms, $\tau_e$ is the erasure constant, and $\beta$ describes the degree of stretch, with $\beta = 1$ corresponding to conventional exponential decay. This stretched-exponential format was previously adopted to describe dark decay[16] but suits us well here because of its similar reciprocity with respect to weakly doped crystals. Source data for Fig. 1 are submitted in Supplementary Data 1.

### Optimization of energy profiles
Data durability is a challenge due to write and read erasure. To manage erasure, we optimize write and read profiles specifying the energy utilized for each IO, which improves both data durability and energy efficiency. As an example since the IO J$^{-1}$ depends on the number of writes and reads done in a zone, we set the number of multiplexed pages in a zone at 400, and sweep the number of reads. Write profile optimization has been studied before[10] that aims for all holograms to have the same diffraction efficiency when all holograms have been written. We adapt this approach by considering the more precise stretched-exponential erasure from our experiments. Our write profile can be obtained via backward-recursion with $\sqrt{\eta_N} = \sqrt{\eta_M} \exp[(\sum_{i=N+1}^{M-1} F_i/\tau_e)^\beta]$, where M is the number of holograms, $\eta_N$ is the diffraction efficiency of the Nth hologram before further holograms being written. $F_i$ is the fluence used to write the i$^{th}$ hologram and can be deduced from Eq. (1). Having swept across various possible diffraction efficiency, best achievable energy efficiency is shown by the solid curve (adapted conventional write optimization) in Fig. 4a.

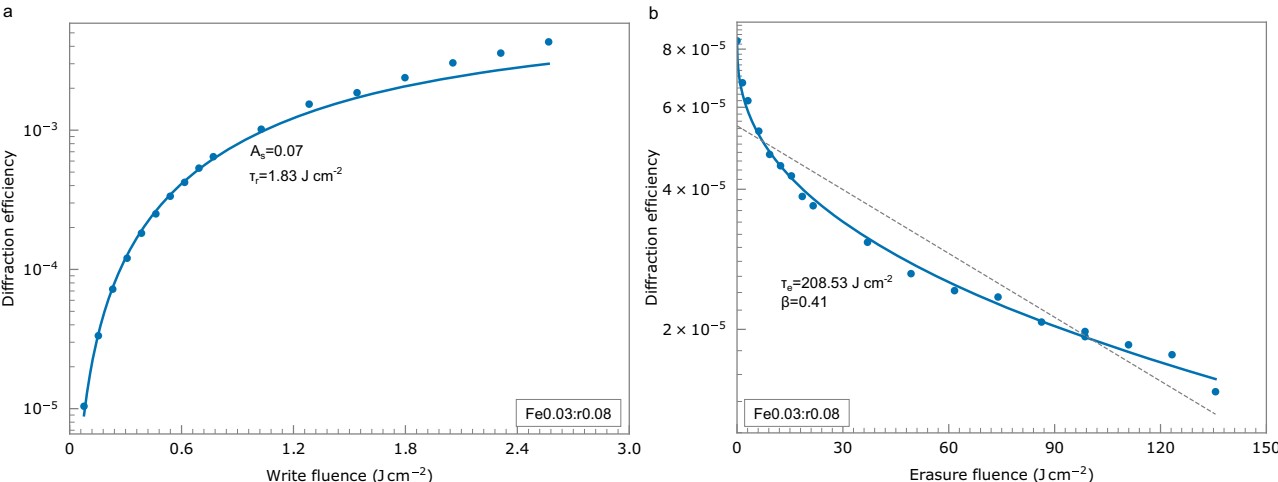

**Fig. 3 | Write and erasure. a** Evolution of diffraction efficiency (log scale) of the first hologram in its write phase. The solid curve shows Eq. (1) fitted to the experimental measurements (dots). **b** Measured (dots) and fitted (solid curve) decay using Eq. (2) of the first hologram (log scale) because of write of later holograms in the erasure phase. Dashed line shows best fit using traditional exponential decal model.

**Fig. 4 | Profile optimization and optimized energy efficiency. a** Energy efficiency and page read gains that can be achieved by jointly optimizing the write and read energy profiles (orange dashed line) for a given multiplexed pages compared to the conventional write optimization (blue solid line). IO stands for input/output.
**b** Optimized write/read energy profiles for workloads with 50% reads with realistic lab constraints. **c** Optimized write/read profiles for workloads with 100% reads. **d** Optimized energy efficiency for workloads with 50% reads achieved from calculations (solid curve) and measurements (cross and square markers). **e** Optimized energy efficiency in log scale for workloads with 100% reads achieved from calculations (solid curve) and measurements (cross markers).

We optimize the read profile by setting the minimum energy incident on camera to a minimum of $e_c$, thereby mitigating the erasure of read holograms. Supplementary Note 2 provides the method of this energy determination process. The read fluence for the $i$th read can be determined from $F_i^R = e_c/(D\eta_i^R)$ with $\eta_i^R$ the diffraction of the holograms after the $N^{th}$ read $\sqrt{\eta_N^R} = \sqrt{\eta_M}\exp[-(\sum_{i=1}^{N} F_i^R/\tau_e)^\beta]$, $D$ is the beam size, and $\eta_M$ the diffraction efficiency of the holograms after the writes. As shown by the dashed curve (joint write and read optimization) in Fig. 4a, the main impact of using variable reads is the possibility to do 1.9× better IO $J^{-1}$ and 3.4× more reads at their best IO $J^{-1}$.

Joint write and read optimization entails tuning the use of varying percentages of the material's dynamic range to optimize for both the targeted number of writes and reads, which depend on the read/write operation ratio inherent in the workload the storage device can handle[17,18]. Consequently, profiles must be optimized according to the read/write ratio. In experiments we implement the above optimization protocol under constraints on the maximum read energy at the media of 10mJ, which is limited by the 2.78 W maximum read power and the maximum write time of 3.6 ms. Figure 4b and c shows selected examples of optimized write and read profiles for 400 multiplexed pages for workloads (assuming a device utilization of 90%) with two extreme read proportions (50% and 100% reads), at $e_c = 30$nJ. As a consequence of the hologram erasure, the energy required to achieve the minimum required diffracted light incident on the camera for successful readout increases with each readout operation. For workloads with 50% reads the number of read operations in a zone before refresh equals the number of writes. Workloads with 100% reads require more write energy to start readout with higher diffraction efficiency and compensate for greater read erasure in order to maximize the number of reads before refresh is required.

After the plotted number of reads is completed, the energy required to readout exceeds 10mJ. At this point we can no longer successfully read out the stored data with the required access rate. Just prior to this point it is necessary to carry out a garbage collection and refresh operation, discarding the deleted pages and rewriting the live pages to a new location before they become unreadable to avoid data losses. To better understand useful Ios, we introduce net IO $J^{-1}$ that refers to the number of user level IO $J^{-1}$ (See Supplementary Note 3 for calculation of net IO $J^{-1}$). This value includes the additional cost of the required system level operations such as garbage collection and refresh. Solid curves in Fig. 4d and e shows the trade-off between energy efficiency (in net IO $J^{-1}$) and number of pages calculated from optimized write and read profiles.

Source data for Fig. 4 are submitted in Supplementary Data 1.

## Demonstrations of energy efficiency, number of reads and density

As an example we experimentally demonstrated the achievable net IO $J^{-1}$ in the crystal Fe0.015:r0.04 via writing, reading and decoding of data pages (and marked the results with cross markers in Fig. 4d, e). For workloads with 50% reads, the calculated optimized net IO $J^{-1}$ was exactly achieved after writing, reading and decoding the data pages (see Supplementary Fig. 3a in Supplementary Note 4 for bit error rate (BER) which relates to user data recovery rate). For workloads with 100% reads, 9.3 K out of 12 K reads had lower BER (Supplementary Fig. 3b in Supplementary Note 4), which equates a smaller net IO $J^{-1}$ as predicted because any uncertainty in the material parameters or experimental energies has big impact on the number of reads and BER that we can achieve. In both experiments, we write 400 multiplexed pages (limited by numerical aperture on the reference beam) with 47KB page size in a volume of 15.3 mm$^3$.

We use the same optimization protocol for a different setup and crystal Fe0.03:r0.08. In this experiment, we increased the page size to 128KB by improving the numerical aperture on the signal beam and moving from 1 bit/symbol to 2 bits/symbol. We increased to 705 multiplexed pages by increasing the reference beam numerical aperture. With data recovery improved by machine learning, we achieved a record net density of 9.6 GB

cm$^{-3}$ in a reduced volume of 5.4 mm$^3$ (see Supplementary Fig. 3c for BER). It's worth noting that with our conventional decoding pipeline, the calculation of net density was not feasible due to excessively high BER. Average write and read times were 13 ms and 2 ms, respectively, resulting in an energy efficiency of 108 raw IO $J^{-1}$ (Supplementary Fig. 3d in Supplementary Note 4).

## Optimization across various Fe:LiNbO3 crystals: measured results and projected trend

We have used framework described above to quantify the best energy efficiency and number of pages across a set of crystals with various doping and annealing conditions to understand the impact of the Fe concentrations on the system performance. We then used the end-to-end optimizer to predict the best achievable performance with an optimally doped and annealed Fe:LiNbO$_3$ Crystal.

Figure 5a shows write efficiency of the first hologram in a fresh crystal as a function of the Fe$^{2+}$ concentration ($c_{Fe2+}$) across all the crystal crystals measured in this work. We find that the write efficiency primarily depends on the Fe$^{2+}$ concentration, with the role of Fe$^{3+}$ concentration being comparatively less impactful. In the lower light regime ( <8.3 W cm$^{-2}$), it increases with increased Fe$^{2+}$ concentrations up to $2.5 \times 10^{17}$cm$^{-3}$ but diminishes at higher Fe$^{2+}$ concentration. Near optimal Fe$^{2+}$ concentrations can be achieved via reducing more Fe$^{3+}$ to Fe$^{2+}$ through post growth annealing. In lowly-doped crystals (e.g. Fe0.001:r6.01), growing crystals with ideal as-grown Fe$^{2+}$ concentration (Fe0.02:r0.09), or oxidising more Fe$^{2+}$ to Fe$^{3+}$ in higher doped crystals (not done with the crystals in this work). The finding that the diffraction efficiency is at the maximum at specific Fe$^{2+}$ concentration is consistent with previous studies[19], but the point at which $A_s/\tau_r$ starts to decrease varies with the recording geometry, crystal size and beam profiles. We expect write efficiency of higher doped crystals with optimal Fe$^{2+}$ concentration will be worse than that of lowly doped crystal with optimal Fe$^{2+}$ because of absorption introduced by more Fe$^{3+}$.

The erasure constant $\tau_e$ is determined primarily by the ratio of Fe$^{2+}$ to Fe$^{3+}$ concentration, as the dielectric relaxation time increases inversely proportional to photoconductivity[3], which was experimentally shown to increase linearly with the concentration ratios[20]. We validated this when writing multiple data pages (Fig. 5b). In crystals with similar Fe$^{2+}$ concentration (e.g. Fe0.001:r6.01 v.s. Fe0.02:r0.09), erasure is slower and more ideal in the higher doped crystal (Fe0.02:r0.09). This is beneficial as less energy is required to compensate for both the write and read erasure, thereby optimizing energy usage.

We calculate the trade-off between energy efficiency and number of pages across various crystals from measured write efficiency and erasure constants (see Fig. 5c for workloads with 50% reads and Fig. 5d for 100% reads). The superior performance of better materials manifests in two dimensions: energy efficiency (vertical axis in Fig. c-d) and the potential for a greater number of pages (horizontal axis). Whist obvious improvement in number of pages is achievable due to the exponential improvement of data durability at lower Fe$^{2+}$ to Fe$^{3+}$ concentration ratios, the gains in energy efficiency are more modest due to the inherent limit on optimal write efficiency. Raising energy efficiency beyond 100 net IO $J^{-1}$ is challenging with a sufficient number of pages. Among the crystals we have measured, Fe0.02:r0.09 exhibits the best performance because its Fe$^{2+}$ concentration approaches the peak of the write efficiency (Fig. 5a), and its Fe$^{2+}$ to Fe$^{3+}$ concentration ratio is sufficiently low to ensure slow erasure (Fig. 5b). At 100 net IO $J^{-1}$, we can potentially multiplex 1482 pages for a workload with 100% reads, and 1918 pages for a workload with 50% reads, corresponding to densities of 4.4 and 5.7 GB cm$^{-3}$ (assuming a small page size of 47kB in a 15.3 mm$^3$ volume).

Finally, we investigate the best performance achievable across untested Fe:LiNbO$_3$. Using 100 net IO $J^{-1}$ as a threshold, we estimate the maximum achievable number of pages at this energy efficiency in relation to the doping level and optimized Fe$^{2+}$ concentration (solid curves in Fig. 5e). (The trend is consistent with the previously shown increase in dynamic range with doping level[21]). Here the media write efficiency and erasure constants are estimated

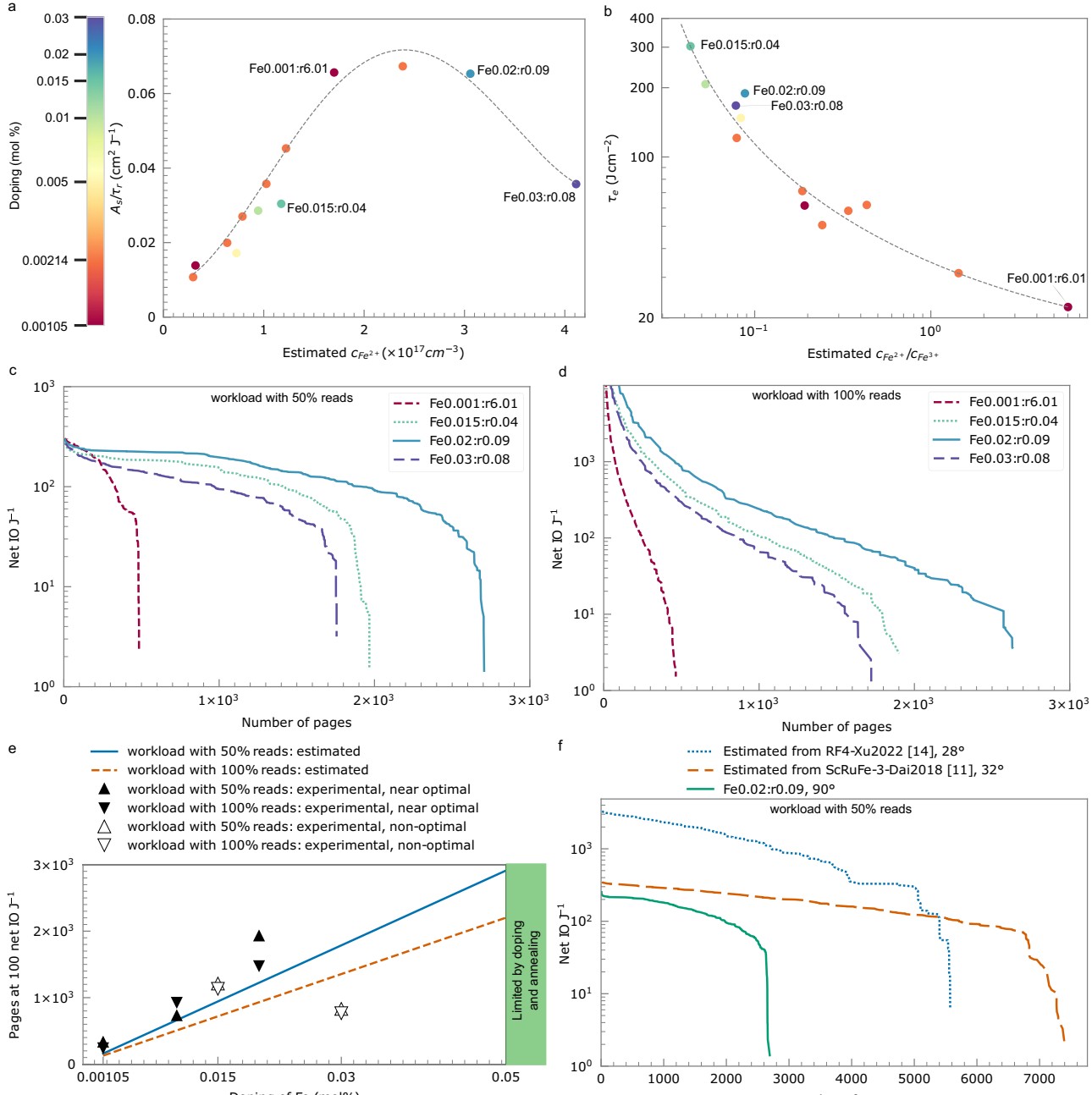

**Fig. 5 | Optimized write, erasure and energy efficiency across various crystals.**
**a** Write efficiency of the first hologram in a fresh crystal as a function of $Fe^{2+}$
concentration. The crystals that will be discussed in the main text are labelled.
Dashed line is a trend line. **b** Erasure constant (log scale) as a function of ratio of $Fe^{2+}$
to $Fe^{3+}$ concentrations (log scale). Dashed curve describes the trend. **c** Net IO $J^{-1}$ (log
scale) at various number of pages calculated from characterized write and erasure
parameters for a workload with 50% reads. IO stands for input/output. **d** Net IO $J^{-1}$
(log scale) at various number of pages calculated from characterized write and
erasure parameters for a workload with 100% reads. **e** Number of pages (log scale)

inferred from characterization experiments (triangles) and estimated number of
pages approximated for 50% and 100% reads (solid and dashed curves) from one-
centre charge transport model in combination with optimal energy search. The
number of pages is threshold at 100 net IO $J^{-1}$ (which corresponds to 27nJ per bit
when writing 47KB data pages). **f** Analysis of optimized energy efficiency (log scale)
for workload with 50% reads in co-doped crystals inferred from literature at
$e_c = 30nJ$. The angles indicated in the legend represent the angles at which the write
and erasure measurements for the materials were conducted.

from a one-centre charge transport model (Supplementary Note 5). For
comparison, measured performance for the subset of crystals that had a $Fe^{2+}$
concentration close to the optimum $2.5 \times 10^{17} cm^{-3}$ are shown as solid
triangles. Our estimates present a conservative trend, as the simplified model
does not account for the degree of stretch β in the erasure procedure that we
have seen in the experiments. The lowly doped crystals are better in
experiments than in estimation because the beneficial stretch exponential is
not considered in the estimation. We also include an example where a higher

doping level, but non-optimal concentrations, results in a lower number of
pages (empty triangles). The experimental data point at 0.03% Fe drops to
lower than estimated optimal because we do not have this crystal sample at
the optimal $Fe^{2+}$ concentration. This result highlights the goal of max-
imizing doping while keeping $Fe^{2+}$ concentration near its optimum.

Our study suggests the potential of writing 2201 pages at 100 net IO $J^{-1}$
for a workload with 100% reads, and 2910 pages for a workload with 50%
reads in a crystal doped at 0.05% Fe and at its optimal oxidation rate.

These number of pages may translate to raw densities of 6.6 and 8.7 GB cm$^{-3}$, assuming a raw page size of 47KB in a 15.3 mm$^3$ volume as we used in media characterization experiments. While the prospect of achieving high density is promising, it remains uncertain whether we can successfully attain the ideal oxidation state in a crystal doped with 0.05% Fe. Moreover, changes in media parameters (e.g. Glass constant) may affect the fidelity of the media's write and erasure characteristics so that they deviate from the current trend.

Our framework also enables performance prediction based on adjustable parameters. For instance, Fig. 5f shows the calculated energy efficiency for workload with 50% reads, considering the write efficiency and erasure constants of co-doped crystals provided by literature[11,14]. In Sc:Ru:Fe:LiNbO$_3$ crystals, the write efficiency, which limits the maximum achievable energy efficiency (to approximately 100 net IO J$^{-1}$) across all Fe:LiNbO$_3$ crystals, could potentially be enhanced. Whilst these results suggest that these materials could provide energy efficiency and density gains, in a holographic storage system we need to caution that the parameters taken from the literature studies were obtained with small angles in a colinear setup and further work is needed to see if these would translate into a 90° setup that is capable of realizing the predicted number of pages. Refer to Supplementary Note 6 for energy efficiency associated with workloads with 100% reads.

Source data for Fig. 5 are submitted in Supplementary Data 1.

## Discussion

To optimize the energy efficiency and density of photorefractive media for warm storage applications, it is essential to balance write, read, and refresh operations. We developed an end-to-end optimizer to determine the optimal properties of Fe:LiNbO$_3$ from a systems perspective. Our model predicts that it is possible to achieve 100 net IO J$^{-1}$ at 2201 pages (for workload with 50% reads), which could correspond to a density of 6.6 GB cm$^{-3}$, assuming a page size of 47KB in 15.3 mm$^3$ volume. The density can further be improved if we increase the page size to 128KB as demonstrated in the density experiment. While the achievable density and energy efficiency defined at the media level are attractive, challenges remain in terms of system loss and system scaling issues when attempting to increase holographic storage to the hundreds of terabytes required to make this cost-effective. Considering only the power consumption of the laser, with a 71% optical system loss and a 15% laser electrical-to-optical efficiency, we can achieve at most 4 IOPS W$^{-1}$. This estimate includes only the energy consumption of the laser, which is expected to be the predominant factor in total costs for our design that features no moving parts. It does not account for cost of other active optical component. To reach a level comparable with HDD, which operates at 20 IOPS W$^{-1}$, we anticipate that a minimum of fivefold improvement in the energy efficiency is needed. Potential upgrades that can enhance both density and energy efficiency performance include more efficient media or a more sensitive camera (see Supplementary Note 7).

## Methods

### Storage hardware rig

Figure 2c shows the storage hardware setup, where we write data bits into the media, recover the output data bits read from the media and measure the performance using both the diffraction efficiency which represents the proportion of reconstructed power to probe power and measures the photorefractive grating strength of the hologram, and the BER. On the signal beam we use a spatial light modulator to encode data, which is coupled into the media using a 2 f system with a Thorlabs TRH254-040-A-ML as the first lens. We write holograms closely after the Fourier plane. To achieve angular multiplexing with a swing of ±14° before the media, we install a scanning mirror and 4 f lens relay system on the reference beam. The angle between axes of the signal beam and the reference beam is set to be 90° to maximize the Bragg selectivity and support the maximum number of angularly multiplexed pages[22]. The axes of expanding signal beam and the reference beam illuminate normally to the crystal surfaces with ordinary polarization. Central axis of the grating vectors is aligned along the polar c axis of the crystal. Illumination of the interference between the two beams on the media

with a controllable power up to 1.4 W realizes change of refractive index and storage of data. To read out the data pages the crystal is illuminated with only the reference beam and the data pages are reconstructed by diffraction of the reference beam from the stored hologram. The diffracted beam is imaged onto an Ximea CB500MG-CM camera.

In the material characterization experiment, we used a Thorlabs EXULUS-4K1/M-SP-SP spatial light modulator configured at 4 × 4 pixels and 1 bit/per symbol, yielding a raw page size of 47KB. The camera lens used was a Nikon AF-S NIKKOR 85 mm f/1.8 G. The reference beam, with a beam waist of 0.28 cm, intersected with the signal beam 4.8 mm away from the focal plane of the signal beam. This intersection resulted in an approximate signal beam size of 0.21×0.26 cm defined at e$^{-2}$.

In the experiment of density demonstration, we used a Meadowlark HSP1920-0532-HSP8 spatial light modulator configured at 2 × 2 pixels and 2bits/symbol, resulting in a raw page size of 128.25 KB. We introduced an aperture onto the reference beam and replaced the camera lens with a Newport PAC070AR, and decreased the volume to dimensions of 1.6 × 2.1 × 2.1 mm.

### Selection of materials

In this work we focus on the characterization of LiNbO$_3$:Fe as this media shows the best write speed among single doped crystals[3]. The Fe concentration is determined from the Fe doping in mol% provided by manufacturers[10], and estimation of Fe$^{2+}$ concentration is based on an absorption measurement at 477 nm with ordinarily polarized light and an assumed absorption cross-section $[Fe^{2+}]\alpha_{477nm} = (2.16 \pm 0.70)\times10^{17}cm^{-2}$ [5,23]. The concentration of Fe$^{2+}$ close to the hologram region was measured, because the concentrations across the 10 × 10 × 20 mm crystals are not uniform due to the technical challenge of uniformly distributing Fe ions in thicker samples[24]. We experimented with crystals having Fe concentrations ranging from $0.2 \times 10^{18}$ to $5.7 \times 10^{18}$ cm$^{-3}$ (Table 1). For those samples with lower doping levels (0.001–0.002% Fe), the ratios of Fe$^{2+}$ to Fe$^{3+}$ exhibited a broad variation.

### Controlled initial condition of experiments

In every experiment, 380 nm ultraviolet (UV) light-emitting diodes (LEDs) are used to erase any refractive index change that was made in previous writes. The erasure process causes the temperature of the crystals increase by up to 24 °C from ambient. To ensure repeatability of the measurements the crystals are let to cool in dark conditions for a sufficient time (15–45 minutes) back to ambient temperature. Every dotted data point shown in this work is achieved in a separate experiment to ensure controlled initial and boundary conditions. The rigs are fully automated to allow experiments to run autonomously and repeatably.

### Selection of write angles and stabilized lasers to enhance reliability

Readout failures may occur (1) consistently with non-ideal write angles, and (2) unrepeatably when the temperature fluctuations exceed the system's tolerance. To enhance reliability, focusing on diffraction efficiency and BER, we (1) identify ideal write angles, and (2) employ lasers with acceptable point stability to accommodate temperature fluctuations.

In an ideal read/write operation, the strongest diffraction efficiency is observed close to the write angle, diminishing at nearby angles in line with Bragg selectivity (the solid curve in Supplementary Fig. 6a in Supplementary Note 8). However, in non-ideal scenarios, the optimal read angle may shift from the write angle (the dashed curve in Supplementary Fig. 6a), or the diffraction efficiency may be reduced (dash dotted curve). Potential causes may include phase coupling between the two writing beams during recording or internal reflections. To mitigate these issues, we refrain from using angles close to 90° in the air and those deviating more than 10° from 90° (unshaded regions in Supplementary Fig. 6b).

To cope with the temperature fluctuations, we employ lasers with acceptable point stability. Utilizing a media characterization setup (Supplementary Fig. 6c) which writes plane wave holograms using a 532 nm Azure Light ALS-GR-532-5-I-CP-SP (pointing stability <0.5 μ °C$^{-1}$) and uses an

additional low power probe beam at 632.8 or 635 nm, we measure the diffraction efficiency of the holograms in real-time and track stability. When we used a <50μrad stabilized laser diode (Edmund optics #33-045) for readout, the peak-to-peak diffraction power variation, based on 29 repetitions of the same experiment over 10 hours, was 15% (Supplementary Fig. 6d). In comparison, employing a basic laser diode (Thorlabs CPS635F) resulted in a higher variation of 112%, where a correlation between diffraction power and temperature was observed. Considering the probabilities of diffraction power, we adjust the read energy to match the lowest anticipated diffraction efficiency when employing Supplementary Fig. 2b. Deviations from the targeted diffraction efficiency are found to increase BER.

## Data availability

The data that support the findings of this study are available from the corresponding author upon reasonable request. The dataset on optimized energy efficiency and density is available at https://doi.org/10.6084/m9.figshare.25723068.v1. Source data for all graphs are submitted in Supplementary Data 1.

## Code availability

The code used to calculate write and read energy profiles is available from the corresponding author upon reasonable request.

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

## Acknowledgements

The authors would like to thank Pashmina Cameron, Joowon Lim, Tony Mason, Marianna Pantouvaki, Soujanya Ponnapalli, Michael Rudow, Theano Stavrinos, Xingbo Wu, Hitesh Ballani, Francesca Parmigiani for valuable discussions.

## Author contributions

B.T. and A.R. conceived the idea and proposed the project. B.T., M.Y., D.K. and J.C. designed the optical setups. M.Y. and J.C. did density related experiments. G.B. and J.C. did material related experiments. N.C. and J.C. worked on material properties modelling and energy efficiency optimization. A.R., N.C. and D.N. worked on system model. B.T. worked on decode of data pages. J.G. worked on machine learning decode of data pages. G.O. and D.K. worked on automation of experiments. G.M. worked on spatial light modulator optimisation. A.S. worked on system efficiency analysis. J.C. wrote the manuscript.

## Competing interests

The authors declare no competing interests.
