## [Peer Review File · Communications Engineering]

Reviewers' comments:

Reviewer #1 (Remarks to the Author):

Holographic data storage is introduced as a replacement for Hard Disk Drives (HDDs) in the cloud. However, the main challenge is data durability due to the erasure which is controlled in iron-doped lithium niobate holographic storage media by means of optimizing iron (Fe) content and annealing process.

I believe that is a high-quality paper offering high potential applications, while encouraging investment in the development of such a holographic data storage to achieve unprecedented data storage density and energy efficiency. Hence, I would recommend publication of this paper. However, I suggest the quality of all figures being improved.

I hope this review letter is beneficial.

Best regards,

Reviewer #2 (Remarks to the Author):

The authors demonstrated a holographic storage system with controllable data erasure. To address the challenge in rewritable holographic media, the authors implemented an end-to-end optimizer for optimal media by balancing read/write/refresh operations, achieving promising density and energy efficiency at the media level. The results are solid and rigorously analyzed. I recommend its publication, subject to the comments below:

- The authors may want to discuss on reliability of the read/write for a fair comparison. It could be failure rate at a certain time span, which is typically used in this community, or any form of estimation that the authors may come up with.
- Can the authors comment on temperature control/thermal management and its impact on holographic performance?
- Regarding the IOPS/W cost, is the cost of the camera (or other optics) included since it looks essential for energy readout?
- A minor point: please double check PBS orientation in Figure 2c right after the input laser.

Reviewer #3 (Remarks to the Author):

Very well written and interesting report.

The authors clearly state it is for cold data storage.

The impact of the machine learning (ML) involved with the purpose to achieve the high reliability and data density is well explained.

Anyway, it would be helpful to show a treated outread pattern if possible and highlight the yield of the ML in an instructive way. This would be a minor revision.

Response to referees

Reviewer #1 (Remarks to the Author):

Holographic data storage is introduced as a replacement for Hard Disk Drives (HDDs) in the cloud. However, the main challenge is data durability due to the erasure which is controlled in iron-doped lithium niobate holographic storage media by means of optimizing iron (Fe) content and annealing process.

I believe that is a high-quality paper offering high potential applications, while encouraging investment in the development of such a holographic data storage to achieve unprecedented data storage density and energy efficiency. Hence, I would recommend publication of this paper. However, I suggest the quality of all figures being improved.

I hope this review letter is beneficial.

Thank you for your feedback. We have updated all figures in the manuscript and supplementary information, aiming to improve the visual appeal and interpretability of all figures, aligning with the journal's guidelines. Our modifications include:

- 1. Adjusting the width of figures in double columns to 18 cm, and those in single columns to 9 cm.*
- 2. Standardizing all figure annotations to use negative integers as unit dimensions, aligning with the manuscript text.*
- 3. Implementing a uniform grey border of the same width for all figures.*
- 4. Minimizing white space in all figures.*
- 5. Eliminating grid lines from all graphs.*
- 6. Adding both major and minor ticks to the graphs.*
- 7. Adopting the recommended optimised colour palettes in figures where colour differentiation aids in distinguishing between data series, in addition to different line patterns.*

Reviewer #2 (Remarks to the Author):

The authors demonstrated a holographic storage system with controllable data erasure. To address the challenge in rewritable holographic media, the authors implemented an end-to-end optimizer for optimal media by balancing read/write/refresh operations, achieving promising density and energy efficiency at the media level. The results are solid and rigorously analyzed. I recommend its publication, subject to the comments below:

- The authors may want to discuss on reliability of the read/write for a fair comparison. It could be failure rate at a certain time span, which is typically used in this community, or any form of estimation that the authors may come up with.

Thank you for your feedback. We detail the reliability observed in our experiments with the addition of a 'Selection of write angles and stabilized lasers to enhance reliability' subsection in the methods section, supplemented by Supplementary Figure 6. In summary, we use BER as a metric to define failure, with the understanding that BER is influenced by diffraction efficiency. To ensure reliable efficiency, we select write and read angles that are characterised to be reliable, and use lasers with good point stability.

- Can the authors comment on temperature control/thermal management and its impact on holographic performance?

Instead of fine temperature control or thermal management, we use lasers with good point stability to improve holographic performance.

In our experiments, we identified one of causes of poor readout performance (low BER) is low diffraction efficiency. One of causes of low diffraction efficiency is a read angle shift, where the optimal read angle diverges from the original write angle. Reading at the original write angle under these circumstances leads to low diffraction efficiency and poor BER. (If we adjust to read from the optimal angle, compensating for the angle shift, data can be recovered).

One factor contributing to read angle shift is temperature variation between the time of reading and when the hologram was written. This issue is dominantly linked to laser point stability in our experiments. As illustrated in the right subplot of Supplementary Figure 6d, we observed a significant change in diffraction efficiency (122% peak-to-peak/mean) correlated with fluctuations in lab temperature. Conversely, as seen in the left subplot of Supplementary Figure 6d, using a different laser with better point stability, under the same lab conditions and with the same setup components, resulted in a much more consistent distribution of diffraction efficiency (15% peak-to-peak/mean) over a span of 10 hours.

- Regarding the IOPS/W cost, is the cost of the camera (or other optics) included since it looks essential for energy readout?

In our current definition, IOPS/W is calculated at the media level, focusing solely on the consumption of laser energy. This is because, in our design, the system has no moving parts, leading us to believe that laser energy is the predominant factor in total operational costs, especially when scaled up. These considerations have been added to the 'Discussion' paragraph.

- A minor point: please double check PBS orientation in Figure 2c right after the input laser.

Thank you for pointing out this detail. We have corrected the PBS orientation.

Reviewer #3 (Remarks to the Author):

Very well written and interesting report.

The authors clearly state it is for cold data storage.

The impact of the machine learning (ML) involved with the purpose to achieve the high reliability and data density is well explained.

Anyway, it would be helpful to show a treated outread pattern if possible and highlight the yield of the ML in an instructive way. This would be a minor revision.

Thank you for your feedback. We added BER decoded from conventional pipeline to Supplementary Figure 3c to show the advantage of ML decoding. ML decoding achieves an acceptable decode rate, leading to the highest net density recorded. In contrast, the excessive BER associated with the conventional pipeline renders the data too error-prone for effective error correction, making net density calculation impossible.

We added a sentence to the first paragraph of the 'Results-demonstrations of energy efficiency, number of reads and density' section, and another to the second paragraph in Supplementary Note 4 to further clarify the contribution of ML decoding.

REVIEWERS' COMMENTS:

Reviewer #1 (Remarks to the Author):

Authors have improved the manuscript based on the comments. I do believe that the current version is ready for the publication.

I hope you find this review helpful.

Best regards,

Reviewer #2 (Remarks to the Author):

The authors addressed all the questions comprehensively and significantly improved the quality of the manuscript. I recommend the publication with its current form.

Reviewer #3 (Remarks to the Author):

The authors have included useful information on how the BER has been improved by the use of machine learning.

Response to referees 24th March, 2024

Reviewer #1 (Remarks to the Author):

Authors have improved the manuscript based on the comments. I do believe that the current version is ready for the publication.

I hope you find this review helpful.

Response: Thank you for your valuable feedback.

Reviewer #2 (Remarks to the Author):

The authors addressed all the questions comprehensively and significantly improved the quality of the manuscript. I recommend the publication with its current form.

Response: The authors appreciate the positive assessment and recommendation for publication. Thank you.

Reviewer #3 (Remarks to the Author):

The authors have included useful information on how the BER has been improved by the use of machine learning.

Response: Thank you for recognizing the enhancements to BER through our application of machine learning.